# Quantifying Uncertainty in the Presence of Distribution Shifts

**Yuli Slavutsky**
Department of Statistics
Columbia University
New York, NY 10027, USA
yuli.slavutsky@columbia.edu

**David M. Blei**
Departments of Statistics, Computer Science
Columbia University
New York, NY 10027, USA
david.blei@columbia.edu

## Abstract

Neural networks make accurate predictions but often fail to provide reliable uncertainty estimates, especially under covariate distribution shifts between training and testing. To address this problem, we propose a Bayesian framework for uncertainty estimation that explicitly accounts for covariate shifts. While conventional approaches rely on fixed priors, the key idea of our method is an adaptive prior, conditioned on both training and new covariates. This prior naturally increases uncertainty for inputs that lie far from the training distribution in regions where predictive performance is likely to degrade. To efficiently approximate the resulting posterior predictive distribution, we employ amortized variational inference. Finally, we construct synthetic environments by drawing small bootstrap samples from the training data, simulating a range of plausible covariate shift using only the original dataset. We evaluate our method on both synthetic and real-world data. It yields substantially improved uncertainty estimates under distribution shifts.

## 1 Introduction

Neural networks are powerful predictive models, capable of capturing complex relationships from data [Raghu et al., 2017]. Despite this capability, they struggle to provide reliable measures of predictive uncertainty. This issue is particularly relevant when the distribution of covariates shifts between training and test data. Such shifts frequently occur in real-world settings, including high-stakes applications like medicine, where inaccurate uncertainty estimates can lead to harmful outcomes [Topol, 2019, Rajkomar et al., 2019].

Consider a dataset $\{(x_i, y_i)\}_{i=1}^n$ and a new test point $x^*$. In a classical Bayesian neural network [Neal, 2012], the posterior predictive distribution is

$$p(y^* \mid x^*, x_{1:n}, y_{1:n}) = \int p(y^* \mid x^*, \theta) \, p(\theta \mid x_{1:n}, y_{1:n}) \, d\theta, \tag{1}$$

where $p(y \mid x^*, \theta)$ comes from the neural network with weights $\theta$ and $p(\theta \mid x_{1:n}, y_{1:n})$ is the posterior over those weights. This expression reveals that, in the classical model, predictive uncertainty arises entirely from uncertainty about model parameters $\theta$. But intuitively, if the new covariate vector $x^*$ lies far from the training covariates $x_{1:n}$, then we should become more uncertain about our prediction.

To capture this intuition, we propose a Bayesian approach that better reflects uncertainty due to covariate shifts. The central idea is to adapt the prior distribution to explicitly depend on covariates, i.e., replacing the classical prior $p(\theta)$ with $p(\theta \mid x_{1:n}, x^*)$. This leads to a posterior predictive distribution of the form

$$p(y^* \mid x^*, x_{1:n}, y_{1:n}) = \int p(y^* \mid x^*, \theta) \, p(\theta \mid x^*, x_{1:n}, y_{1:n}) \, d\theta. \tag{2}$$

39th Conference on Neural Information Processing Systems (NeurIPS 2025).

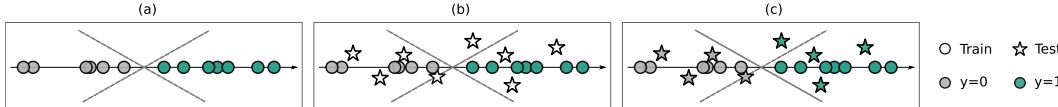

Figure 1: (a) In training one covariate is fixed; the data lies on a one-dimensional subspace. All predictors intersecting the fixed axis at the same point are equivalent. (b) At test time, variation along the second dimension reveals that some predictors may fit better the new data, prompting a prior shift. (c) Possible labeling where only the solid line separates the test data.

As we will discuss, conditioning the prior on the test covariates $x^*$ allows the posterior to adjust its uncertainty in accordance with the proximity of $x^*$ to the training distribution. Thus, the model captures the impact of covariate shift on predictive performance, and delivers increased uncertainty for inputs that lie far from the training data.

The intuition behind this prior is that predictions become more uncertain at covariates far from the training data because the learned relationship may no longer hold. Consider a logistic regression with two covariates, where one covariate varies substantially and the other hardly varies. The classical posterior of the coefficients can assign both low and high value to the less variable covariate since in the training data it is almost indistinguishable from the intercept. But if the test data includes a previously unseen value of this covariate, our predictive uncertainty should increase. The test data point, even without its response, indicates that the coefficient could differ significantly from what the training data alone suggests. For an illustration see Figure 1.

Implementing this idea presents three key challenges:

The first challenge is to specify the prior $p(\theta \mid x_{1:n}, x^*)$. We propose an energy-based prior that spreads its mass on the plausible values of the weights, given the covariates.

The second challenge is to compute Equation (2), which involves the posterior distribution $p(\theta \mid x^*, x_{1:n}, y_{1:n})$. Unlike a classical posterior distribution, this posterior is located in the context of a prediction about $x^*$, which comes into the adaptive prior. To approximate it, we use the idea of *amortized variational inference* [Kingma et al., 2015, Margossian and Blei, 2024]. We learn a family of approximate posteriors that take test covariates $x^*$ as input and produce an approximate posterior tailored to its prediction.

A final challenge is that fitting our amortized variational family requires both training data from the training distribution and new data from a covariate-shifted distribution. In practice, however, we often only have one training set, without access to data drawn from a shifted distribution. We use small bootstrap samples to form synthetic environments [Slavutsky and Benjamini, 2024] and prove that they contain covariate-shifted distributions able to approximate unseen shifts. We adapt the variational objective to match all of these environments.

Together, these ideas form Variational Inference under Distribution Shif (VIDS). On both real and synthetic data, we show that VIDS outperforms existing methods in terms of predictive accuracy, calibration of uncertainty, and robustness under covariate shifts. VIDS provides accurate estimates of posterior predictive uncertainty in the face of distribution shift.

**Related work.**     Forming predictions under covariate shift is important to many applications. Examples from the medical domain include imaging data from different hospitals [Zech et al., 2018, AlBadawy et al., 2018, Perone et al., 2019, Castro et al., 2020], and failure to provide reliable predictions when applied to different cell types. In image classification, cross-dataset generalization remains challenging [Torralba and Efros, 2011], including cases where shifts are introduced by variations in cameras [Beery et al., 2018], or by temporal and geographic differences. Other examples include person re-identification, where the training data often includes biases with respect to to gender [Grother et al., 2019, Klare et al., 2012], age [Best-Rowden and Jain, 2017, Michalski et al., 2018, Srinivas et al., 2019], and race [Raji and Buolamwini, 2019, Wang et al., 2019]. All of these settings that can benefit benefit from the use of VIDS.

Approaches for uncertainty estimation in neural networks include ensemble-based methods [Lakshminarayanan et al., 2017, Valdenegro-Toro, 2019], and methods based on noise addition [Dusenberry et al., 2020, Maddox et al., 2019, Wen et al., 2020]. A prominent line of work focuses on Bayesian neural networks [Tishby et al., 1989, Denker and LeCun, 1990], which offer a principled framework for uncertainty quantification and have been widely adopted for this purpose [Ovadia et al., 2019]. These approaches (e.g., [Ha et al., 2016, Yoon et al., 2018]) typically place a single prior distribution

over the model parameters that is shared across all inputs. In contrast, our method defines a prior that adapts to individual covariates.

Several techniques have been given Bayesian interpretations, including regularization methods such as dropout [Kingma et al., 2015, Gal and Ghahramani, 2016], stochastic gradient-based approximations [Welling and Teh, 2011, Dubey et al., 2016, Li et al., 2016], and variational inference methods that approximate the posterior distribution [Graves, 2011, Neal, 2012, Blundell et al., 2015, Louizos and Welling, 2016, Malinin and Gales, 2018].

A prevalent strategy for handling uncertainty under distribution shift involves distance-aware methods, which estimate uncertainty based on the distance between new inputs and the training data. These include approaches that rely on training data density estimation [Sensoy et al., 2018], often implemented via kernel density methods [Ramalho and Miranda, 2020, Van Amersfoort et al., 2020] or Gaussian Processes (GPs) [Williams and Rasmussen, 2006]. More recent examples include Spectral-normalized Neural Gaussian Processes (SNGP) [Liu et al., 2020], which apply spectral normalization to stabilize network weights, and Deterministic Uncertainty Estimation (DUE) [Van Amersfoort et al., 2021], which integrates a GP with a deep feature extractor trained to preserve distance information in the representation space.

Most relevant to our work are methods that combine Bayesian and distance-based approaches, such as Park and Blei [2024], which incorporate an energy-based criterion into the training objective to increase predictive uncertainty for inputs that are unlikely under the training distribution.

The key idea behind the aforementioned distance-based approaches is to detect shifts by measuring how far test inputs lie from the training inputs, typically using fixed metrics in the input space. But these measures can be misleading because not all shifts affect predictive performance equally. Consider a univariate logistic regression. Suppose there are two types of shifts: (i) training data concentrate near the decision boundary, and (ii) training data appear only at extreme covariate values. At test time, covariates cover the entire range. Under shift (i), predictive performance remains strong because the critical region near the boundary was covered during training. Under shift (ii), performance deteriorates because the model never saw data near the decision boundary.

The important factor is not simply the difference in covariates, but how shifts influence predictive accuracy. Unlike predefined distance measures, the adaptive prior in VIDS essentially learns from data how covariate shifts affect predictive performance. It uses its adaptive prior to directly model the change in predictive uncertainty that is induced by newly observed covariates.

## 2   Predictive uncertainty under distribution shifts

We address uncertainty estimation under covariate shift, where the distribution of test-time inputs $x^*$ may differ from that of the training data $x_{1:N}$. To account for such shifts, we extend the classical Bayesian framework by *treating covariates as random variables and modeling their dependence on the model parameters $\theta$*. This leads to a formulation in which the prior over $\theta$ is conditioned on the newly observed covariate $x^*$, resulting in a predictive posterior that explicitly reflects this dependence. Consequently, the predictive uncertainty, defined through this posterior, can adapt to reflect greater uncertainty for covariates $x^*$ that are unlikely under the training distribution, thereby capturing their potential impact on predictive performance. We begin by revisiting the classical Bayesian model.

### 2.1   Background on the classical Bayesian model

In the classical Bayesian framework the parameters $\theta$ are treated as a random variable drawn from a prior distribution $\theta \sim p(\theta)$, while the covariates $x_{1:N}$ are considered fixed. Under the standard conditional independence assumption, each outcome $y_i$ is independent of all other pairs $(x_j, y_j)$ given $\theta$ and $x_i$. The posterior distribution over $\theta$ is

$$p(\theta|x_{1:N}, y_{1:N}) \propto p(\theta) \prod_{i=1}^{N} p(y_i|x_i, \theta). \tag{3}$$

New test points $x^*$ are likewise treated as fixed. Thus, the introduction of a new test input $x^*$ does not alter the posterior, and consequently does not affect the predictive uncertainty. Again, see Equation 1.

## 2.2 A Bayesian approach to covariate shift

Our goal in this work is to explicitly model distributional differences between the training covariates $x_1, \ldots, x_N \sim p_x$ and a new covariate $x^* \sim p_{x^*}$. To effectively capture how such shifts impact predictive performance, the posterior predictive at $x^*$ should adapt to the observed change in the covariate distribution. To this end, we propose a model in which the model parameters $\theta$ depend on both the training and test covariates, $x_{1:N}$ and $x^*$. This dependence is illustrated in the probabilistic graphical model in Figure 2, which indicates the structure of the posterior predictive for $x^*$.

As in the classical framework, our model assumes that $y^*$ is conditionally independent of all other variables given $x^*$ and $\theta$. However, by allowing $\theta$ to depend on the covariates, the prior $p(\theta|x_{1:N}, x^*)$ can now *adjust the plausibility of parameter values based both on training and test inputs, and thus capture distribution shifts*. As a result, the posterior $p(\theta|x_{1:N}, y_{1:N}, x^*)$ is explicitly dependent on $x^*$ through the covariate-dependent prior.

Let $f_\theta$ denote the predictive model parametrized by $\theta$. The predictive uncertainty for $x^*$ in our model is defined through its predictive posterior in Equation 2, which is our primary quantity of interest. Our **goal** is to approximate this predictive uncertainty. Since the likelihood $p(y^* \mid x^*, \theta)$ can be evaluated directly by the predictive model $f_\theta(x^*)$, the central challenge lies in accurately estimating the covariate-dependent posterior $p(\theta|x^*, x_{1:N}, y_{1:N})$.

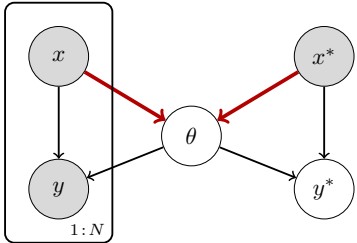

In what follows, we first define a concrete adaptive prior $p(\theta|x^*, x_{1:N})$ that conditions on both the observed training covariates and the test-time covariate. We then develop a variational inference scheme to approximate the posterior under this adaptive prior.

Figure 2: Graphical model. Thick red arrows denote additional dependencies introduced by our model. Observed variables shown in gray.

Finally, since test-time covariates $x^*$ from a shifted distribution are typically unavailable during training, we approximate this setting by constructing *synthetic environments* designed to simulate diverse covariate distributions by subsampling the training data. Using these environments, we design an algorithm that approximates a posterior capable of anticipating predictive degradation under a range of potential test-time shifts.

## 2.3 The adaptive prior

Our prior aims to capture the plausibility of $\theta$ given both the training inputs $x_{1:N}$ and a new test covariate $x^*$. We propose an adaptive prior conditioned on both $x_{1:N}$ and $x^*$, defined by the following energy function

$$E(\theta; x_{1:N}, x^*) := \int \sum_{i=1}^{N} \log p(y|x_i, \theta) + \log p(y|x^*, \theta) \, dy \qquad (4)$$

$$p(\theta|x_{1:N}, x^*) := \frac{1}{Z(\theta)} \exp\left(E(\theta; x_{1:N}, x^*)\right), \qquad (5)$$

where $Z(\theta) := \int \exp\left(E(\theta; x_{1:N}, x^*)\right) d\theta$ is the normalizing factor.[1]

This formulation allows for a smooth adaptation to test-time shifts. When only a small number of test covariates $x^*$ are introduced, or if the inputs are similar to the training data, the prior remains close to a distribution conditioned on the training covariates alone. However, as more test inputs are observed, especially if they differ substantially from the training distribution, the prior adjusts more significantly to reflect the new covariate distribution.

We illustrate this adaptivity in the simple example described in §1. We consider a logistic model for $y \in \{0, 1\}$, where covariates $x$ consist of two features: one of them remains constant in all training examples, but in new test examples, both features vary. Figure 3 shows how the prior distribution changes when the shifted examples are introduced. See Appendix A for additional details and an illustration that lightly shifted examples lead to minimal adaptations.

---

[1]This definition requires integrability of $\exp\left(E(\theta; x_{1:N}, x^*)\right)$, and thus we assume that $\lim_{\|\theta\| \to \infty} E(\theta; x_{1:N}, x^*) = -\infty$ with at least linear decay.

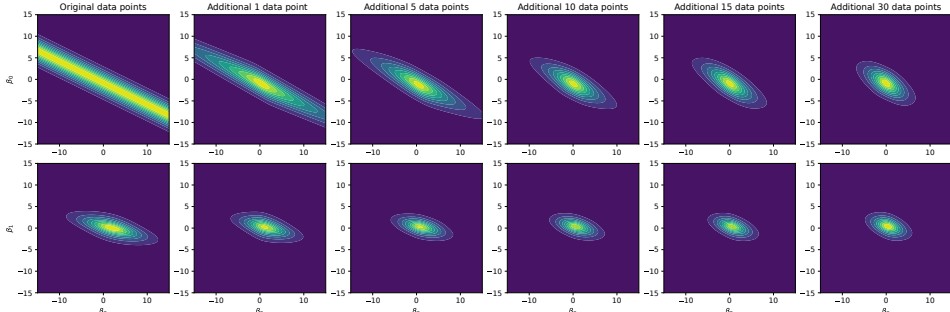

Figure 3: Changes in the prior due to the introduction of test covariates drawn from a shifted distribution $x^* \sim \mathcal{N}(\frac{1}{2}, 1)$, where both features vary.

## 2.4 VIDS: Variational inference under distribution shifts

Given this prior, our goal is to estimate the posterior $p(\theta \mid x_{1:N}, y_{1:N}, x^*)$, which enables approximate computation of predictive uncertainty for the test input $x^*$ according to Equation (2). For this, we fit a variational distribution $q_\phi(\theta; x^*) \approx p(\theta \mid x_{1:N}, y_{1:N}, x^*)$. This variational distribution is an *amortized posterior approximation*, parametrized as a function of $x^*$, allowing approximation of the posterior across multiple test-time covariates $x^*$.

We model the amortized posterior as a multivariate Gaussian distribution, parametrized by its mean $\mu$ and diagonal covariance matrix $\Sigma$. Let $\mathcal{Q}_d$ denote the family of $d$-dimensional Gaussian distributions with diagonal covariance. Thus, we seek $q_\phi(\theta; x^*)$ with $\phi = (\mu, \Sigma)$, that minimizes the Kullback-Leibler divergence to the true posterior:

$$\min_{q_\phi \in \mathcal{Q}_d} \mathrm{KL}\left(q_\phi(\theta; x^*) \,\|\, p\left(\theta \mid x_{1:N}, y_{1:N}, x^*\right)\right). \tag{6}$$

Specifically, we optimize the evidence lower bound (ELBO) on the log-likelihood [Blei et al., 2017, Kingma and Welling, 2014, Rezende and Mohamed, 2015]

$$\mathcal{L}\left(\phi; x^*, \mathcal{D}\right) = \mathbb{E}_{q_\phi}\left[\log p\left(y_{1:N}|x_{1:N}, \theta\right)\right] - \mathrm{KL}\left(q_\phi(\theta; x^*) \,\|\, p\left(\theta|x_{1:N}, x^*\right)\right), \tag{7}$$

which is equivalent to solving Equation (6).

We train a neural network $h_\gamma$ with weights $\gamma$, to output $\phi$ from $x^*$, and optimize $\gamma$ rather than $\phi$ directly. Given the training set $\mathcal{D} = \{(x_i, y_i)\}_{i=1}^N$ and $M$ test covariates $\{x_j^*\}_{j=1}^M$, we define the following objective to fit our amortized posterior

$$\mathcal{L}_\mathcal{D}(\gamma) = \sum_{j=1}^M \mathcal{L}\left(\phi_j; x_j^*, \mathcal{D}\right) = \sum_{j=1}^M \mathcal{L}\left(h_\gamma(x_j^*); x_j^*, \mathcal{D}\right). \tag{8}$$

Note that evaluation of this objective requires estimation of the prior

$$p(\theta|x_{1:N}, x^*), \tag{9}$$

which involves integration over the outcome space $\mathcal{Y}$. For discrete $\mathcal{Y}$ the integration is simply summation, and thus can be easily computed. If $\mathcal{Y}$ is continuous, we apply Monte Carlo integration: we sample $r$ target values uniformly from an integration range $[y_{\min}, y_{\max}]$, and for each sample compute the log-likelihood under a unit-variance Gaussian centered at each predicted value.

## 2.5 The variational family and stochastic optimization of the variational objective

To fully describe VIDS, it remains to define the amortized variational family and the optimization procedure to maximize Equation 8. So far, the variational posterior has been amortized with respect to $x^*$. Now, we will amortize also with respect to the training set, in order to better optimize across multiple environments, as described in the next section.

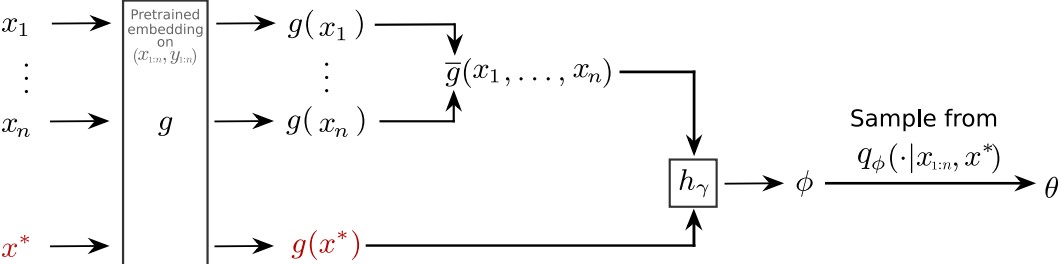

Figure 4: Optimization mechanism for a single test example in a single synthetic environment.

The variational family is $q_\phi(\theta; x^*, x_{1:n})$ where the parameters $\phi$ come from an inference network. We decompose the inference network into an embedding network $g : \mathbb{R}^d \to \mathbb{R}^k$, parametrized by $\xi$, and a prediction layer parametrized by $\theta$. We assume that $g$ has been pre-trained to maximize the likelihood $p(y|x, \theta) = f_\theta(g_\xi(x))$ on the training dataset. We focus on the prediction layer $\theta$.

Given the training data $x_{1:n}$ and test covariate $x^*$, we define the inference network as follows:

1. We compute the embeddings of the training set, $\hat{g}(x_1), \ldots, \hat{g}(x_n)$, and aggregate them into a single summary statistic (e.g., the mean)[2] denoted by $\overline{g}(x_1, \ldots, x_n)$.

2. We compute the embedding of the test covariate, $\hat{g}(x^*)$.

3. The aggregated training embedding and the test embedding are concatenated and passed through a network $h_\gamma : \mathbb{R}^{2k} \to \Phi$, which outputs the parameters $\phi = (\mu, \Sigma)$ of the variational distribution $q_\phi(\theta \mid x_{1:n}, x^*)$.

In practice, we optimize a single-sample Monte Carlo estimate of the expectations in Equation 7. To enable differentiable sampling from $q_\phi$, we use the *reparametrization trick* [Kingma and Welling, 2014], where we sample $\theta \sim q_\phi$ by first sampling $\epsilon$ from a standard Gaussian and then calculating $\theta = \mu + \Sigma\epsilon$. The full procedure is summarized in Algorithm 1 and illustrated in Figure 4.

### 2.6 Uncertainty estimation

With training completed using Algorithm 1, we finally turn to estimating predictive uncertainty for a new test input $x^*$, i.e., from the posterior predictive distribution of Equation (2).

Given the learned parameters $\hat{\gamma}$, we evaluate the representation of the test point $g_{\hat{\xi}}(x^*)$ and use it with the combined representation of the training data to compute the variational posterior parameters via $\hat{\phi} = h_{\hat{\gamma}}(\overline{g}_{\hat{\xi}}(x_{1:N})$. We then approximate the posterior predictive by drawing samples $\theta^{(1)}, \ldots, \theta^{(S)} \sim q_{\hat{\phi}}$ and calculating the corresponding predictions $f_{\theta^{(s)}}(g_{\hat{\xi}}(x^*))$.

## 3 Posterior estimation across multiple distribution shifts

So far, we have developed a method to approximate the posterior given training data $\mathcal{D} = \{(x_i, y_i)\}_{i=1}^N$ and a set of test covariates $\{x_j^*\}_{j=1}^M$, drawn from a shifted distribution. However, a key challenge in real-world settings is that such test covariates are typically unavailable in advance. To address this limitation, similarly to [Slavutsky and Benjamini, 2024], we generate *synthetic environments* via subsampling from the training data.

Specifically, we construct $L$ environments by sampling $L$ pairs of datasets

$$\mathcal{D}_{\text{tr}}^{(\ell)} = \{(x_1^{(\ell)}, y_1^{(\ell)}), \ldots, (x_n^{(\ell)}, y_n^{(\ell)})\}, \quad \mathcal{D}_{\text{te}}^{(\ell)} = \{(x_1^{*(\ell)}, y_1^{*(\ell)}), \ldots, (x_m^{*(\ell)}, y_m^{*(\ell)})\}, \quad (10)$$

where each dataset pair is constructed by sampling data pairs $(x_i, y_i)$ uniformly at random with replacement from the original training set $\mathcal{D}$. We refer to each resulting pair of datasets as a synthetic

---

[2]The summary embedding $\overline{g}(x_1, \ldots, x_n)$ is a permutation-invariant aggregation of learned element-wise representations, and thus follows the Deep Sets setting [Zaheer et al., 2017].

---

**Algorithm 1** Variational posterior

---

1: **Input:** Training data $\mathcal{D}$, covariates $\{x_j^*\}_{j=1}^M$, pre-trained embedding $g_{\hat{\xi}}$, predictor $f(\cdot; \theta)$, iterations $K$, learning rate $\eta$, initialization $\gamma^{(0)}$.

2: Compute train embeddings $g_{\hat{\xi}}(x_1), \ldots g_{\hat{\xi}}(x_N)$ and test embeddings $g_{\hat{\xi}}(x_1^*), \ldots, g_{\hat{\xi}}(x_M^*)$

3: Aggregate train embeddings to obtain $\overline{g}_{\hat{\xi}}(x_1, \ldots, x_N)$

4: **for** $1 \leq k \leq K$ **do**

5:    **for** $1 \leq j \leq M$ **do**

6:       Compute $\phi_j^{(k)} = h(\overline{g}_{\hat{\xi}}(x_{1:N}), g_{\hat{\xi}}(x_j^*); \gamma^{(k-1)})$

7:       Sample $\epsilon_j^{(k)}$ and compute $\theta_j^{(k)} = \mu_j^{(k)} + \Sigma_j^{(k)} \cdot \epsilon_j^{(k)}$ for $(\mu_j^{(k)}, \Sigma_j^{(k)}) = \phi_j^{(k)}$

8:       Compute

$$p(y_{1:N} \mid x_{1:N}, \theta_j^{(k)}) = \{f_{\theta_j^{(k)}}(g_{\hat{\xi}}(x_i))\}_{i=1}^N, \quad p(y_j^* \mid x_j^*, \theta_j^{(k)}) = f_{\theta_j^{(k)}}(g_{\hat{\xi}}(x_j^*)),$$
$$p(\theta_j^{(k)} \mid x_{1:N}, x_j^*) \text{ (prior; Equation 5)}, \qquad q_{\phi_j}(\theta_j^{(k)} \mid x_{1:N}, y_{1:N}, x_j^*)$$

9:    **end for**

10:   Compute $\mathcal{L}^{(k)} = \sum_{j=1}^m \mathcal{L}_{\mathcal{D}}(\phi_{1:m}^{(k)})$

11:   Update the parameters of $h$ by performing a gradient ascent step:

$$\gamma^{(k)} \leftarrow \gamma^{(k-1)} + \eta \nabla_\gamma \mathcal{L}_{\mathcal{D}}^{(k)}$$

12: **end for**

    **Return:** $\hat{\gamma} := \gamma^{(K)}$

---

environment, denoted $e^{(\ell)} = \left\{\mathcal{D}_{\text{tr}}^{(\ell)}, \mathcal{D}_{\text{te}}^{(\ell)}\right\}$. Each such synthetic test set $\mathcal{D}_{\text{te}}^{(\ell)}$ is likely to exhibit a different empirical distribution, thereby simulating a potential covariate shift.

The core idea of our approach can viewed as an inverse bootstrap sampling: while bootstrap sampling relies on subsamples of the original dataset being highly likely to resemble the population distribution, we instead focus on the low-probability cases where the subsample deviates from the original dataset's distribution. These deviations simulate potential distribution shifts that may arise at test time.

In the following proposition we show that drawing enough subsamples guarantees that with high probability, at least one of them will be close to the true unknown test distribution.

**Proposition (informal) 3.1.** *Let $p$ and $p^*$ be binned distributions of the training data and the unobserved test set, respectively. Assume that $\text{supp}(p^*) \subseteq \text{supp}(p)$. Then, for any $\epsilon > 0$ and $0 \leq \alpha < 1$, there exist $m$ and $L$ such that, with probability at least $1 - \alpha$, the empirical distribution of at least one of $L$ randomly drawn subsamples of size $m$ from the training data satisfies $\|\hat{p}^{(\ell)} - p^*\|_1 \leq \epsilon$.*

Appendix B gives a formal proposition, the proof, an application to the case where $\text{supp } p^* \not\subseteq \text{supp } p$, and analysis of relationships between the number of required synthetic environments $L$ to $\epsilon$ and $\alpha$.

While the proposition guarantees that, given enough sampled environments, at least one will approximate the true (unknown) test-time shift, it remains unclear which one that is. To address this, we aim to ensure that the learned posterior performs well across all synthetic environments. This motivates the use of environment-level penalties, inspired by the out-of-distribution (OOD) generalization literature [Arjovsky et al., 2019, Wald et al., 2021, Krueger et al., 2021].

Thus, we introduce the following cross-environment objective:

$$\mathcal{L}^{(\ell)} = \sum_{x^* \in \mathcal{D}_{\text{te}}^{(\ell)}} \mathcal{L}_{\mathcal{D}_{\text{tr}}^{(\ell)}}(\phi_{1:m}^{(\ell)}; x^*) \tag{11}$$

$$\mathcal{L} = \sum_{\ell=1}^L \mathcal{L}^{(\ell)} + \tau \, \text{Var}\left(\mathcal{L}^{(1)}, \ldots, \mathcal{L}^{(L)}\right). \tag{12}$$

Here, we set the penalty to the variance across the environments, as proposed by Krueger et al. [2021]. In Algorithm 2 we summarize the complete procedure of variational posterior estimation with synthetic environments (see Appendix C).

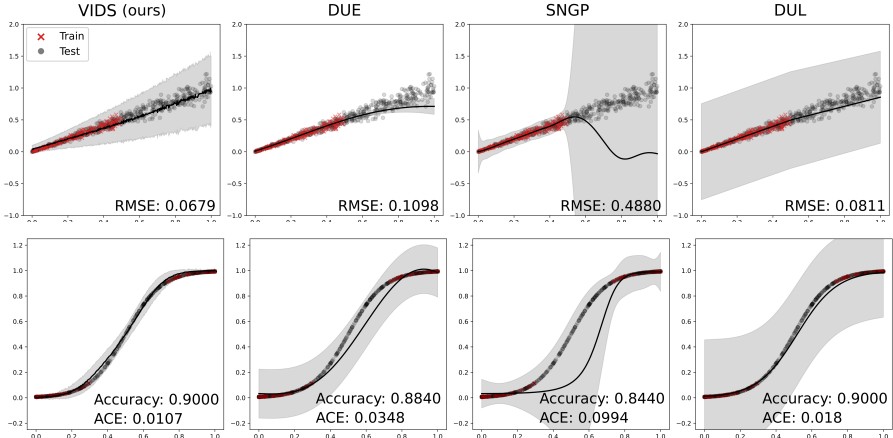

Figure 5: Simulation results. Red crosses represent training data, and gray dots test data. Black lines depict predictions, gray shaded area spanning $\pm 1$ standard deviation. Top: Heteroskedastic linear regression for $a = 0.5$; Bottom: Binary classification with missing data for $t = 0.3$. VIDS is the only one to capture correct variance structures and thus achieves the best results.

## 4 Experiments

We evaluate VIDS on both synthetic and real-world datasets, across classification and regression tasks. We compare the uncertainty estimates produced by VIDS (ours) with previous distance aware methods: SNGP [Liu et al., 2020], DUE [Van Amersfoort et al., 2021], and distance uncertainty layers (DUL) [Park and Blei, 2024] (see §1 for more details). In all experiments, the same neural network architecture is used as the prediction model. Hyper-parameters of our and competing methods were optimized via grid search to maximize average performance (accuracy for classification, RMSE for regression) on a single sample of $J = 50$ synthetic test environments of size $m = 10$, which was discarded from the analysis. For the corresponding values, and additional implementation details see Appendix E.

### 4.1 Synthetic data

We begin by examining two synthetic examples exhibiting covariate shifts. Each experiment uses $N = M = 500$ training and test points, and VIDS constructs synthetic environments of size $m = 20$. All models use a fully connected neural network with one hidden layer of width $d = 8$.

**4.1.1 Regression: heteroscedastic linear model** We sample $x \sim \mathcal{U}[0, a]$, and $x^* \sim \mathcal{U}[0, b]$ for $a < b$. Outcomes for train and test follow $y = \beta x + \epsilon(x)$ where $\epsilon(x) \sim \mathcal{N}(0, \frac{x}{10})$. Results for $a = 0.5, b = 1$ and $\beta = 1$ are shown in figure 5, and results for additional settings in Appendix D.

*Results:* VIDS consistently achieves the lowest RMSE and is the only one to capture the correct variance structure. DUE exhibits low variance but underestimates uncertainty at higher $x$ values. SNGP overfits, yielding poor test performance and excessive variance. DUL provides the best posterior mean among competitors, but overestimates uncertainty due to incorrect variance modeling.

**4.1.2 Classification: logistic regression with missing data** For both the training and test sets, we sample $x \sim \text{Beta}(1/2, 1/2)$ and compute $\rho(x) = \sigma(-5 + 10x)$ where $\sigma$ denotes the standard sigmoid function. Outcomes are then generated according to $y|x \sim \text{Ber}(\rho(x))$. However, in the training data, we exclude middle values within the range $(t, 1 - t)$. Results for $t = 0.3$ are shown in Figure 5.

*Results:* VIDS achieves the highest accuracy and lowest calibration error. In addition, as can be seen in Figure 5, VIDS exhibits lowest variance while correctly modeling uncertainty: higher in the unseen middle and lower at the edges. DUE and, to a greater extent, DUL predict well but misrepresent uncertainty, increasing variance at the edges, with DUL overestimating variance overall. SNGP estimates higher variance in the middle but consistently under-predicts.

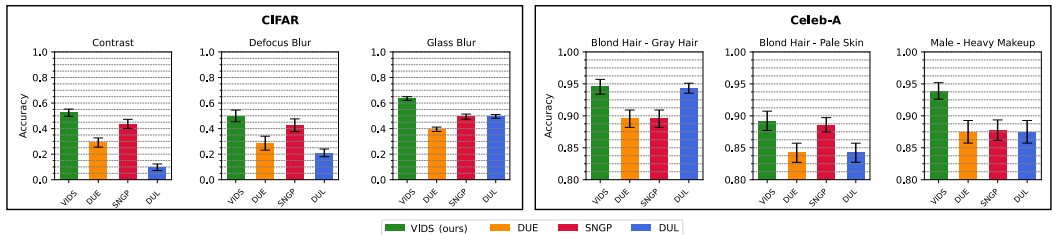

Figure 6: Classification accuracy over 10 repetitions. Celeb-A titles formatted as Target – Shift Attr. VIDS achieves highest accuracy in all experiments.

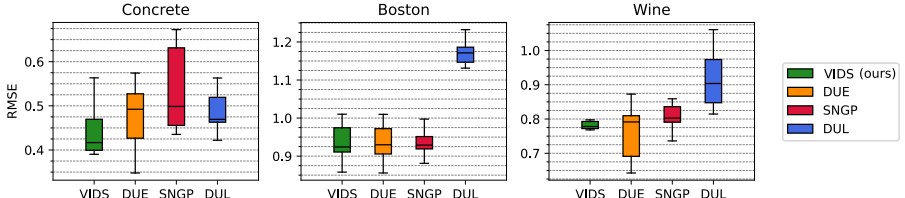

Figure 7: RMSE results for three experiments on three UCI regression datasets over 10 repetitions. VIDS achieves lowest or comparable RMSE in all experiments.

## 4.2 Real data

### 4.2.1 Classification

For the classification experiments we use a simple convolutional neural-network with two convolution blocks (see details in Appendix E).

**Corrupted CIFAR-10** We evaluate model performance under corruption-induced distribution shifts using the CIFAR-10-C dataset [Hendrycks and Dietterich, 2019]. We perform experiments on three corruption types: defocus blur, glass blur, and contrast. We construct the training set of 5000 images, 90% clean from the original CIFAR-10 dataset [Krizhevsky et al., 2009] and 10% corrupted images from CIFAR-10-C, while the test set is constructed with 5000 images, 90% corrupted and 10% clean.

**Celeb-A** For each experiment, we choose one annotated attribute as the target, and another attribute $A$ to induce a distribution shift in the CelebA dataset [Liu et al., 2015]. The training set contains 500 images with 90% having $A = 1$ and 10% with $A = 0$; the test set reverses this ratio: 90% images with $A = 0$ and 10% with $A = 1$. We run three such experiments with the following shift–target pairs: (i) Pale Skin $\rightarrow$ Blond Hair, (ii) Heavy Makeup $\rightarrow$ Male, and (iii) Gray Hair $\rightarrow$ Blond Hair.

*Results:* Figure 6 shows that in all 6 classification experiments VIDS achieves better accuracy.

### 4.2.2 Regression

We conduct experiments on three UCI regression datasets—Boston, Concrete, and Wine. For each, we designate a prediction target and apply K-Means clustering ($K = 2$) on all numeric covariates. To simulate a non-trivial covariate shift, we use the cluster with the higher average within-cluster Euclidean distance primarily for training (90% of training data), and the lower-distance cluster primarily for testing (90% of test data). In all these experiments we use a simple linear regression (one-layer network) as the base model. For additional details see Appendix E.

*Results:* Figure 7 shows that VIDS achieves the lowest average RMSE across all three datasets and consistently low variance, while DUE under-performs on the Concrete and Boston datasets and DUL considerably under-performs on the Wine dataset.

## 5 Conclusion

We introduced VIDS, a Bayesian method for quantifying uncertainty under potential distribution shifts. VIDS leverages two central ideas: incorporating dependencies between network parameters and the test covariates, and balancing performance across synthetic environments to simulate covariate shifts. VIDS consistently outperforms existing methods, particularly when assumptions like homoscedasticity or continuity are violated.

## 6 Acknowledgments

We are grateful to members of the Blei Lab for fruitful discussions and feedback. In particular, to Sebastian Salazar, Eli N. Weinstein, Andrew Jesson, Nicolas Beltran, and Sweta Karlekar. We thank Andrew Jesson for verifying the DUE implementation. This work is supported by NSF IIS-2127869, NSF DMS-2311108, ONR N000142412243, the Simons Foundation and DoD OUSD (R&E) under Cooperative Agreement PHY-2229929 (The NSF AI Institute for Artificial and Natural Intelligence). YS is supported by a Founder's Postdoctoral Fellowship, Department of Statistics, Columbia University.

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

# A  Example for the adaptive prior

We revisit the simple example from §1, considering a binary prediction setting where $y \in \{0, 1\}$ and the covariates $x \in \mathbb{R}^2$ consist of two features, $x^{(1)}$ and $x^{(2)}$. In the training data $x^{(2)}$ remains constant across all examples. However, at test time, new subtypes may be encountered where both features exhibit variability.

Assume the following logistic model

$$\rho(x) = \beta_0 + \beta_1 x^{(1)} + \beta_2 x^{(1)}, \quad p(y|x) = \sigma(\rho(x)), \tag{13}$$

where $\sigma(x) = 1/(1 + e^{-\rho(x)})$ and $\theta = (\beta_0, \beta_1, \beta_2)^{\mathsf{T}}$.

Due to the exchangeability between $\beta_0$ and $\beta_2$ in the training data, all combinations with a fixed value of $\beta_0 + \beta_2$ are equally plausible when observing only the training data. However, this symmetry is broken when test data from new subtypes is observed.

As a concrete example, assume that for all datapoints (train and test) the first feature is distributed as $x^{(1)} \sim \mathcal{N}(1, 1)$. In the training data, $x_i^{(2)} = \frac{1}{2}$ for all $1 \leq i \leq N$; however, for new test examples $x^*$, the second feature is drawn from $\mathcal{N}\left(\frac{1}{2}, 1\right)$. Let $\theta' := (\beta_2, \beta_1, \beta_0)^{\mathsf{T}}$. While for any of the first $N$ datapoints, $p(y|x_i, \theta) = p(y|x_i, \theta')$, this no longer holds for $x^*$, thus leading to changes in the prior distribution upon arrival of a new test example $x^*$.

Figure 3 illustrates how the prior distribution, which for $y \in \{0, 1\}$ corresponds to

$$p(\theta|x_{1:N}) \propto \exp\left(\sum_{i=1}^{N} \log p(y = 0|x_i, \theta) + \sum_{i=1}^{N} \log p(y = 1|x_i, \theta)\right), \tag{14}$$

adapts when new test examples are observed.

For the training data, any combination of $\beta_0$ and $\beta_2$ with a fixed sum $\beta_0 + \beta_2$ results in the same probability. However, as new examples with varying values of $x^{(2)}$ are introduced, the prior distribution begins to depend on how the weight is distributed between $\beta_0$ and $\beta_2$. Consequently, the equivalence regions concentrate around specific combinations of $\beta_0$ and $\beta_2$.

If, instead, the second feature of the test examples $x^*$, is drawn from a distribution closely aligned with the training data, $\mathcal{N}\left(\frac{1}{2}, \frac{1}{10^2}\right)$, i.e., a distribution closely similar to the training data, the resulting changes to the prior are minimal. This is illustrated in Figure 8.

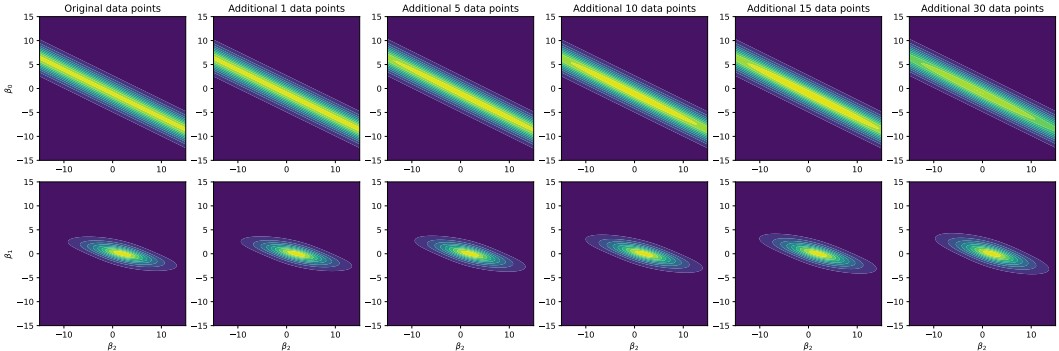

Figure 8: Changes in the prior due to introduction of test covariates drawn from a distribution similar to the training data, $x^* \sim \mathcal{N}(\frac{1}{2}, \frac{1}{10^2})$. In the top row, $\beta_0$ is fixed at $-1$, while in the bottom row, $\beta_1$ is fixed at $1$.

# B    Theoretical analysis of synthetic-environment sampling

The statement of Proposition 3.1 holds for any pair of finite distributions $p, p^* \in \Delta^k$, not just those arising from our specific application—namely, the empirical distribution of the training set and the binned test distribution. In what follows, we formally state and prove this more general result, and then return to its application in our setting.

**Proposition B.1.** *Let $B_1, \ldots, B_k$ be a partition of $\mathcal{X}$. Denote the binned empirical distribution of the train set $x_{1:N}$ as $p \in \Delta^k$, where $p(i) = \frac{1}{N} \sum_{j=1}^{N} \mathbb{1}\{x_j \in B_i\} \forall 1 \leq i \leq k$. Similarly, define the binned test distribution induced by the partition as $p^*(i) = \int_{B_i} p_{x^*}(x)\, dx$. Assume that $\operatorname{supp} p^* \subseteq \operatorname{supp} p$. Then for any $\epsilon > 0$ and $\alpha \in (0, 1)$, there exist integers $m, L \in \mathbb{N}$, such that if $L$ independent samples of size $m$ are drawn according to $p$, then with probability at least $1 - \alpha$, at least one sample will induce a binned empirical distribution $\hat{p}^{(\ell)}$ satisfying $\|\hat{p}^{(\ell)} - p^*\|_1 \leq \epsilon$.*

*Proof.* Let $m := \lceil \frac{2(k-1)}{\epsilon} \rceil$ and define $q \in \Delta^k$ as follows

$$q(i) := \begin{cases} \frac{\lfloor m\, p^*(i) \rfloor}{m}, & 1 \leq i \leq k-1 \\ 1 - \sum_{i'=1}^{k-1} \frac{\lfloor m\, p^*(i') \rfloor}{m}, & i = k. \end{cases} \tag{15}$$

For $1 \leq i \leq k-1$, we have $|q(i) - p^*(i)| \leq \frac{1}{m}$, and $|q(k) - p^*(k)| \leq \frac{k-1}{m}$. Hence, $\|q - p^*\|_1 \leq \frac{2(k-1)}{m} \leq \epsilon$.

Define

$$T = \{z = z_{1:m}\, ;\, p_z = q\} \tag{16}$$

the set of all samples of size $m$ drawn according to $p$, such that their empirical distribution $p_z = q$. This is the type-set of $q$. Denote the probability of drawing a given sample $z = z_{1:m}$ by $p^m(z) := \prod_{j=1}^{m} p(z_j)$. Then, by the method of types (see, e.g., Theorem 11.1.4 in [42]), the probability of drawing a sample whose empirical distribution equals $q$ is

$$p^m(T) = \sum_{z \in T} p^m(z) \geq \frac{1}{(m+1)^k} e^{-m\, \mathrm{KL}(q\|p)} =: \xi. \tag{17}$$

Consequently, the probability of drawing $L$ samples of size $m$ according to $p$, such that the empirical distribution of at least one of them is $q$ satisfies

$$1 - \left(1 - p^m(T)\right)^L \geq 1 - (1 - \xi)^L. \tag{18}$$

Taking $L \geq \frac{\log \alpha}{\log(1-\xi)}$ completes the proof. $\square$

**Remark B.2.** Using the inequality $\log(1 - \xi) \geq -\frac{\xi}{1-\xi}$, we have

$$\frac{\log \alpha}{\log(1-\xi)} \leq \frac{1-\xi}{\xi} \log \frac{1}{\alpha} = \left((m+1)^k e^{m\, \mathrm{KL}(q\|p)} - 1\right) \log \frac{1}{\alpha} \tag{19}$$

$$\leq \left(\left(\frac{2(k-1)}{\epsilon} + 2\right)^k e^{\left(\frac{2(k-1)}{\epsilon} + 1\right) \mathrm{KL}(q\|p)} - 1\right) \log \frac{1}{\alpha} = O\left(\left(\frac{1}{\epsilon}\right)^k e^{\frac{1}{\epsilon}} \log \frac{1}{\alpha}\right), \tag{20}$$

revealing the relationship of $L$ to the tolerance $\epsilon$ and to $\alpha$.

**Application to our setting**    We now return to the setting in which $p \in \Delta^k$ denotes the binned empirical distribution of the train set $x_{1:N}$, induced by the partition of $\mathcal{X}$. Specifically,

$$p(i) = \frac{1}{N} \sum_{j=1}^{N} \mathbb{1}\{x_j \in B_i\} \tag{21}$$

for all $1 \leq i \leq k$. Similarly, let $p^* \in \Delta^k$ denote the binned test distribution induced by the partition, given by

$$p^*(i) = \int_{B_i} p_{x^*}(x)\, dx. \tag{22}$$

If $\operatorname{supp} p^* \subseteq \operatorname{supp} p$, then the conditions of Proposition 3.1 are satisfied, and thus drawing enough subsamples from $x_{1:N}$ guarantees that, with high probability, in the binned space at least one of them will be close to the test distribution.

However, if there exists a bin $B_i$, such that $x_j \notin B_i$ for all $1 \leq j \leq N$ but $\int_{B_i} p_{x^*}(x)\,dx > 0$, then if $\|p - p^*\|_1 = \epsilon' < \epsilon$, by discarding any bins $B_i$ such that $B_i \in \operatorname{supp} p^*$ and $B_i \notin \operatorname{supp} p$, and re-normalizing the probabilities on the remaining bins to sum to 1, we obtain a reduced distribution for which we can require a distance to $p^*$ that is at most $\epsilon - \epsilon'$. Consequently, Proposition 3.1 can be applied directly to this re-normalized distribution, guaranteeing with high probability that at least one subsample will be close to the test distribution within the reduced tolerance $\epsilon - \epsilon'$.

## C    Multi-environment algorithm

---

**Algorithm 2** Variational posterior with synthetic environments

---

1: **Input:** Data $\mathcal{D}$, no. synthetic environments $L$, train and test environment sizes $n$ and $m$, pre-trained embedding $g_{\hat{\xi}}$, predictor $f(\cdot;\theta)$, iterations $K$, learning rate $\eta$, initialization $\gamma^{(0)}$.

2: **for** $1 \leq k \leq K$ **do**
3:     **for** $1 \leq \ell \leq L$ **do**
4:         Sample synthetic datasets: $\mathcal{D}_{\text{tr}}^{(\ell)} = \{(x_i^{(\ell)}, y_i^{(\ell)})\}_{i=1}^n, \quad \mathcal{D}_{\text{te}}^{(\ell)} = \{(x_j^{*(\ell)}, y_j^{*(\ell)})\}_{j=1}^m$
5:         Compute and test embeddings: $g_{\hat{\xi}}(x_1^{(\ell)}), \ldots g_{\hat{\xi}}(x_n^{(\ell)})$ and $g_{\hat{\xi}}(x_1^{*(\ell)}), \ldots, g_{\hat{\xi}}(x_m^{*(\ell)})$
6:         Aggregate train embeddings to obtain $\overline{g}_{\hat{\xi}}(x_1^{(\ell)}, \ldots, x_n^{(\ell)})$
7:         **for** $1 \leq j \leq m$ **do**
8:             Compute $\phi_j^{(k,\ell)} = h(\overline{g}_{\hat{\xi}}(x_{1:n}^{(\ell)}), g_{\hat{\xi}}(x_j^{*(\ell)}); \gamma^{(k-1)})$
9:             Sample $\epsilon_j^{(k,\ell)}$ and compute $\theta_j^{(k,\ell)} = \mu_j^{(k,\ell)} + \Sigma_j^{(k,\ell)} \cdot \epsilon_j^{(k,\ell)}$ for $(\mu_j^{(k,\ell)}, \Sigma_j^{(k,\ell)}) = \phi_j^{(k,\ell)}$
10:           Compute:

$$p(y_{1:n}^{(\ell)} \mid x_{1:n}^{(\ell)}, \theta_j^{(k,\ell)}) = \{f_{\theta_j^{(k,\ell)}}(g_{\hat{\xi}}(x_i^{(\ell)}))\}_{i=1}^n, \quad p(y_j^{*(\ell)} \mid x_j^{*(\ell)}, \theta_j^{(k,\ell)}) = f_{\theta_j^{(k,\ell)}}(g_{\hat{\xi}}(x_j^{*(\ell)})),$$
$$p(\theta_j^{(k,\ell)} \mid x_{1:n}^{(\ell)}, x_j^{*(\ell)}) \text{ (prior; Eq. 5)}, \qquad q_{\phi_j^{(k,\ell)}}(\theta_j^{(k,\ell)} \mid x_{1:n}^{(\ell)}, y_{1:n}^{(\ell)}, x_j^{*(\ell)})$$

11:         **end for**
12:     **end for**
13:     Compute $\mathcal{L}^{(k)} = \frac{1}{\sum} \sum_{\ell=1}^L \sum_{j=1}^m \mathcal{L}_{\mathcal{D}}(\phi_{1:m}^{(k,\ell)})$
14:     Update the parameters of $h$ by gradient ascent: $\gamma^{(k)} \leftarrow \gamma^{(k-1)} + \eta \nabla_\gamma \mathcal{L}^{(k)}$
15: **end for**

---

## D    Additional experimental results

### D.1    Synthetic data

Full results for the synthetic experiments presented in Figure 5 are provided in Table 1.

Table 1: Results for synthetic data linear regression and binary classification

| Experiment | Metric | VIDS (ours) | DUE | SNGP | DUL |
|---|---|---|---|---|---|
| **Linear** | RMSE | **0.068** (0.002) | 0.110 (0.008) | 0.488 (0.004) | 0.081 (0.010) |
| **Logistic** | Accuracy | **0.900** (0.009) | 0.880 (0.001) | 0.844 (0.027) | **0.900** (0.008) |
| | ACE | **0.011** (0.003) | 0.034 (0.007) | 0.099 (0.038) | 0.018 (0.000) |

In Figure 9 we provide results for the heteroscedastic regression experiment, for additional values of the parameter $a$, controlling how much the test data is shifted with respect to the observed training data, and thus the difficulty of generalization on providing accurate uncertainty measures.

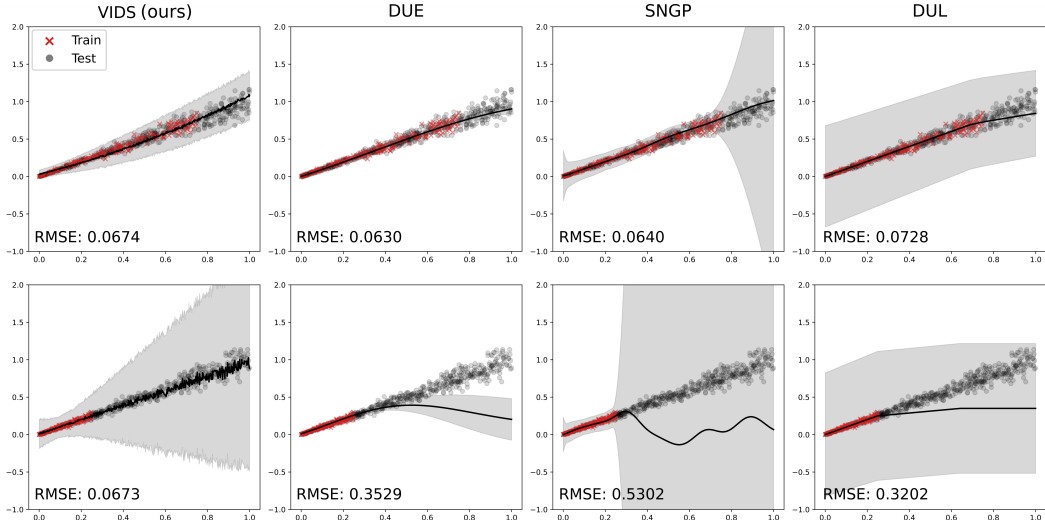

Figure 9: Simulation results for heteroskedastic regression for $a = 0.25$ (top) and $a = 0.75$ (bottom). Red crosses represent training data, while gray dots indicate test data. The black line depicts the predictions, with the gray shaded area spanning $\pm 1$ standard deviation. A single repetition is depicted; RMSE values are averaged over 10 repetitions.

# E   Implementation Details

## E.1   General implementation details

**Code**   The code to reproduce our results is attached to the submission and upon acceptance a link to a permanent repository will be included in the main text.

All the code in this work was implemented in Python 3.11. We used Numpy 2.0, TensorFlow 2.13 and TensorFlow Addons 0.21 packages. The UCI datasets were loaded through sklearn 1.6. CIFAR-C dataset was obtained from Zenodo[3]. CIFAR-10 and Celeb-A datasets were loaded through torchvision 0.21. All figures were generated using Matplotlib 3.10.

For DUE and DUL implementation was adapted from the source code of the original papers[45]. For SNGP implementation was adapted from the implementation provided in the source code of DUE.

We ran all synthetic data and UCI experiments on 2 CPUs. Each repetition of these experiments lasted less than 7 minutes. For real data classification experiments (on CIFAR-10 and Celeb-A datasets) we used a single A100 cloud GPU. Each repetition lasted less than 18 minutes.

**Hyper-parameters**   Our setting deals with an unknown covariate shift. Thus, hyperparameters for all methods were chosen via a grid-search in a single experiment repetition, (excluded from the analysis). Some of the hyperparameters are method specific. Our variational method uses environment related parameters: number of environments $J$, size of each test environment $m$, and size of each train environment $n$. Below we refer to the KL penalty $\lambda$ as 'penalty', and to the variance penalty as $\tau$. The DUE method employs inducing points to approximate the Gaussian process component. Both DUE and SNGP use GP features, random Fourier features used for approximating the kernel, and scaling of the input features. In DUL the number of steps corresponds to epochs. The hyperparameters of all methods are specified in the corresponding tables below.

**Normalizing factor**   For continuous response variables, our implementation approximates the log of the normalizing factor $z$ using the log of the mean of exponential log-likelihoods: $\log v \approx \log(\frac{1}{n} \sim_i \exp(v_i))$ where $v_i$ are the integrated log likelihoods.

---

[3] https://zenodo.org/record/2535967/files/CIFAR-10-C.tar
[4] https://github.com/y0ast/DUE
[5] https://github.com/yookoon/density_uncertainty_layers

Table 2: Hyperparameters for heteroskedastic linear and logistic regression.

| Parameter | Linear | | | | Logistic | | | |
|---|---|---|---|---|---|---|---|---|
| | VIDS (ours) | DUE | SNGP | DUL | VIDS (ours) | DUE | SNGP | DUL |
| $J$ | 30 | – | – | – | 30 | – | – | – |
| $m$ | 20 | – | – | – | 20 | – | – | – |
| $n$ | 500 | – | – | – | 500 | – | – | – |
| $\tau$ | 0.001 | – | – | – | 0.001 | – | – | – |
| Penalty | 0.005 | – | 1 | 1 | 0.005 | – | 1 | 1 |
| Batch size | 520 | 100 | 64 | 50 | 520 | 100 | 32 | 50 |
| Steps | 30 | 1500 | 7000 | 500 | 30 | 1500 | 1000 | 300 |
| Learning rate | $10^{-2}$ | $10^{-2}$ | $10^{-3}$ | $10^{-3}$ | $10^{-3}$ | $10^{-2}$ | $10^{-3}$ | $10^{-2}$ |
| n_inducing_points | – | 20 | – | – | – | 20 | – | – |
| GP features | – | – | 128 | – | – | – | 3 | – |
| Random features | – | – | 1024 | – | – | – | 128 | – |
| Feature scale | – | – | 2 | – | – | – | 2 | – |

## E.2 Implementation details for synthetic data experiments

The data-related parameters of the synthetic experiments are described in the main text. The hyper-parameters used for the heteroskedastic linear regression and logistic regression with missing data are detailed in Table 2.

For VIDS we specify $h_\gamma$ as a fully-connected neural network with 6 layers of sizes $64d, 32d, 16d, 8d, 4d, 2d$ and ReLU activation between the layers, for $d = 8$.

## E.3 Implementation details for real data classification experiments

For the classification experiments (both for CIFAR and Celeb-A datasets) and all methods, we specify the base model as a convolutional neural network with two convolutional blocks, each with a 3×3 convolution with 32 filters, followed by a ReLU activation and a 2×2 max-pooling. These are followed by a fully connected layer with 64 units, ReLU activation and a final fully connected layer of dimension $d = 16$.

We performed a grid search for hyperparameters for each method on a single repetition of the experiment on an excluded setting. We chose the "Pixelate" corruption for the search on CIFAR, and the Target – Shift Attribute pair of Male – Blurry for Celeb-A. The resulting hyper-parameters are reported in Table 3.

**CIFAR** For VIDS, we specify $h_\gamma$ as a fully connected neural network with 4 layers of sizes layers of sizes $32d \cdot 10, 16d \cdot 10, 4d \cdot 10, 2d \cdot 10$ and ReLU activations between them, for $d = 16$.

**Celeb-A** For VIDS, we specify $h_\gamma$ as a fully connected neural network with 5 layers of sizes $32d, 16d, 8d, 4d, 2d$ and ReLU activations between them, for $d = 16$.

## E.4 Implementation details for real data regression experiments

We use the standard target variables from the UCI datasets: MEDV for Boston, Compressive Strength for Concrete, and Quality for Wine.

To evaluate model performance under distribution shifts, we split each dataset into two groups by applying the K-Means algorithm with $K = 2$ on all numerical columns, excluding the target variable. We calculate the average Euclidean within-cluster distance for each cluster as the mean distance from each point to its centroid. The cluster with the larger average within-cluster distance is designated as the majority in training.

We then sample data from both clusters uniformly at random to form the training and test sets. The training set contains 90% of the high-variance cluster, while the test set consists of 90% of the

Table 3: Hyperparameters for classification experiments.

| Parameter | CIFAR | | | | Celeb-A | | | |
|---|---|---|---|---|---|---|---|---|
| | VIDS (ours) | DUE | SNGP | DUL | VIDS (ours) | DUE | SNGP | DUL |
| $J$ | 10 | – | – | – | 10 | – | – | – |
| $m$ | 20 | – | – | – | 20 | – | – | – |
| $n$ | 5000 | – | – | – | 1000 | – | – | – |
| $\tau$ | 0.001 | – | – | – | 0.001 | – | – | – |
| Penalty | 0.001 | – | 1 | 1 | 0.005 | – | 1 | 1 |
| Batch size | 5020 | 100 | 100 | 50 | 1020 | 100 | 100 | 50 |
| Steps | 25 | 25,000 | 100,000 | 500 | 50 | 1500 | 10,000 | 500 |
| Learning rate | $10^{-4}$ | $10^{-4}$ | $10^{-4}$ | $10^{-3}$ | $10^{-3}$ | $10^{-2}$ | $10^{-3}$ | $10^{-3}$ |
| n_inducing_points | – | 100 | – | – | – | 20 | – | – |
| GP features | – | – | 64 | – | – | – | 16 | – |
| Random features | – | – | 512 | – | – | – | 64 | – |
| Feature scale | – | – | 2 | – | – | – | 2 | – |

low-variance cluster. Both the training and test features and target variables are standardized using the median and standard deviation of the training set.

For all experiments we specified the base model as a fully connected neural network with 4 layers of sizes $8d, 4d, 2d, 4d, 2d$ and ReLU activations between them, for $d = 32$.

The hyper-parameters for the experiments are detailed in Table 4.

Table 4: Hyperparameters for UCI regression experiments.

| Parameter | VIDS (ours) | DUE | SNGP | DUL |
|---|---|---|---|---|
| $J$ | 10 | – | – | – |
| $m$ | 50 | – | – | – |
| $n$ | 1000 | – | – | – |
| $\tau$ | 0.001 | – | – | – |
| Penalty | 0.01 | – | 1 | 1 |
| Batch size | 520 | 100 | 64 | 50 |
| Steps | 150 | 1000 | 7000 | 500 |
| Learning rate | $10^{-3}$ | $10^{-2}$ | $10^{-3}$ | $10^{-3}$ |
| n_inducing_points | – | 20 | – | – |
| GP features | – | – | 16 | – |
| Random features | – | – | 16 | – |
| Feature scale | – | – | 2 | – |

## E.5  Running times

Our method differs from others in that it pre-trains a representation and performs variational inference only on the prediction layer. As a result, for smaller models (e.g., linear regression), our approach can be more computationally expensive due to per-environment optimization. However, for larger models, our method has an advantage since competing methods must optimize many more parameters. In the following table, we specify the hardware and execution times for all methods across our experiments.

Table 5: Runtime comparison across methods.

| Dataset | Hardware | Unit | Vids | DUE | SNGP | DUL |
|---|---|---|---|---|---|---|
| Linear regression | 2 CPUs | seconds | 67.160 | 39.900 | 54.800 | 16.400 |
| Logistic regression | 2 CPUs | seconds | 61.250 | 40.300 | 16.500 | 6.290 |
| Concrete | 2 CPUs | minutes | 3.237 | 2.070 | 0.700 | 0.400 |
| Boston | 2 CPUs | minutes | 3.112 | 2.017 | 0.717 | 0.267 |
| Wine | 2 CPUs | minutes | 2.523 | 1.583 | 1.567 | 0.895 |
| CIFAR-C (max across corruptions) | A100 Cloud GPU | minutes | 2.516 | 7.447 | 3.527 | 4.040 |
| Celeb-A (max across tasks) | A100 Cloud GPU | minutes | 2.420 | 4.417 | 4.220 | 3.318 |

