# OpenReview forum: "Quantifying Uncertainty in the Presence of Distribution Shifts"
_NeurIPS.cc/2025/Conference — NeurIPS 2025 poster_

### Official Review · Reviewer_xBus · 2025-06-17

**Clarity:** 2
**Significance:** 2
**Originality:** 2
**Rating:** 5
**Confidence:** 3

**Summary:**

The paper proposes a Bayesian approach where the prior distribution is adapted to additionally depend on test covariates. The objective is to enable the model to better capture covariate shift (from training data to test data), thus improving predictive performance and uncertainty calibration. To facilitate application, amortized variational inference and synthetic environments are used. Experiments on classification and regression task are reported.

**Questions:**

1. As stated above: Is the submission truly the first work to develop the idea of (additionally) conditioning on test covariates, as is suggested? What is the relation to e.g. conditional neural processes (e.g., Garnelo et al. 2018), Bayesian Model-Agnostic Meta-Learning (Yoon et al. 2018) and, e.g., other work on amortized variational inference / variational hypernetworks?
2. Why was ACE used as evaluation metric instead of, e.g., ECE?
3. Can you provide some more information on the runtime of the method (elaborating on the information in Appendix E)?

If question 1 is addressed satisfactorily, I would be open to increasing my evaluation score, especially if question 2 and 3 are also addressed.

**Ethical Concerns:**

["NO or VERY MINOR ethics concerns only"]

**Final Justification:**

The quality, clarity and significance of the paper had been strong from the beginning. However, I had been unsure about the originality of the work. The authors’ clarifications during the discussion phase were able to resolve my concerns in this regard. I also find the responses to the other questions to be satisfactory and appreciate the authors’ suggestions to incorporate them into the camera ready version. Thus, I overall recommend to accept the paper.

**Limitations:**

yes (although their severity could be elaborated on)

**Paper Formatting Concerns:**

I have not noticed any major formatting issues in this paper.

**Quality:**

3

**Strengths And Weaknesses:**

Quality: Overall, I think the paper is a quality submission. I did not identify a technical issue within the paper. The code is available, as is a theoretical analysis in the Appendix. While the experiments could be extended (e.g., larger/more complex datasets), results on both classification and regression are presented and implementation details for the experiments are also provided. Discussion on strengths and limitations is included in a brief manner, but could also be extended (e.g., how severe are the currently discussed limitations?).

Clarity: The beginning of the paper was quite clear and straightforward to read, with the rationale clearly outlined. I then found the later parts of Section 2 to be quite hard to follow, especially the technicals/math. I suggest to cut some redundancies from the beginning of Section 2, which are already presented in a similar fashion in Section 1, and to provide more explanations on 2.3-2.5, e.g., the energy function, amortized variational inference and the sampling strategies. Some smaller suggestions for clarity: The use of the ACE metric is not justified. “Selection bias” is highlighted in the Abstract, but not really discussed further in the paper. Also, the references to the example in “§1” (i.e., Section 1) are not clear.

Significance: Quantifying uncertainty in machine learning is undoubtedly of enormous importance. I thus believe that the paper aims at an important contribution, and the results of the experiments are strong enough. That being said, I am unsure about the originality of the work, which also affects its significance (cf. below).

Originality: Overall, I believe that the justification of the paper’s originality needs improvement. The idea of (additionally) conditioning on test covariates is a good one, but is the submission truly the first work to develop this idea, as is suggested? For example, I am not sure about the relation of the paper to conditional neural processes (e.g., Garnelo et al. 2018): Don’t these effectively also incorporate a conditioning on test inputs? I have similar concerns regarding Bayesian Model-Agnostic Meta-Learning (Yoon et al. 2018) and, e.g., other work on amortized variational inference / variational hypernetworks. The paper must be clearer in this regard, i.e., what truly is novel. Additionally, I found the presentation of related work in the paper to be suboptimal. There is a bolded term “Related work” in Section 1, which, however, partly reads more like a “Background” section (e.g., discussing examples from the medical domain). Why not dedicate a separate section to important related approaches and clarify the contribution and novelty of the paper? Moreover, almost all of the work discussed in the “related work” is from 2017-2021. Has there hardly been any progress since then?

I wish the authors all the best for their work on this interesting topic.

References:
Garnelo, M., Rosenbaum, D., Maddison, C., Ramalho, T., Saxton, D., Shanahan, M., ... & Eslami, S. A. (2018, July). Conditional neural processes. In International conference on machine learning (pp. 1704-1713). PMLR.

Yoon, J., Kim, T., Dia, O., Kim, S., Bengio, Y., & Ahn, S. (2018). Bayesian model-agnostic meta-learning. Advances in neural information processing systems, 31.

---

> ### Author Rebuttal · Authors · 2025-07-30
>
> Dear reviewer,
>
> Thank you for your thoughtful review and your clarity regarding score reevaluation. Please see our responses.
>
> **Question 1:**
>
> To the best of our knowledge, this is indeed the first work to introduce a covariate-dependent prior. Below we briefly review the literature you mentioned and highlight key differences from our approach with respect to the prior treatment. We will address these references in the paper as well.
>
> 1. _Conditional Neural Processes (Garnelo et al. 2018)_
>
>     This paper addresses the problem of predicting outputs for a new set of inputs (the target set) given a set of input-output pairs (the context set). The model uses an encoder to map each labeled pair $(x_i, y_i)$ in the context set to a representation $r_i$. These representations are then aggregated into a summary vector $r$, which, together with a new target input $x^*$ is passed to the predictor to produce the output.
>
>     _Key difference_ : The proposed model is trained by maximizing the conditional likelihood, following a purely frequentist approach with **no prior**. Conditioning occurs only with respect to training covariates (and labels)  and the model remains fixed afterward. **New covariates are treated as standard inputs** to the prediction function.  In contrast, our model is fully Bayesian and incorporates an **adaptive prior**: not only does it produce a different posterior for each new covariate $x^ * $, but the **prior itself changes** with $x^ * $. This dependence is learned by optimizing a variational objective, where both the prior and posterior are explicitly conditioned on $x^ * $.
>
> 2. _Bayesian Model-Agnostic Meta-Learning (Yoon et al. 2018)_
>
>     The paper aims to learn a parameter initialization $\theta$ such that a few gradient updates on a new task $T_j$ lead to good performance. To capture task-specific dependencies, it places a particle-based prior over the task-specific parameters.
>
>     _Key difference_: The particle-prior is $\{\theta_1, … , \theta_m\}$, corresponding to tasks  $T_1, …, T_m$.  The prior is not covariate specific. It uses training data for different tasks, but **does not depend on individual covariates**.
>
> 3. _Hypernetworks_ have a similar dependence structure: a hyper-network $H$ generates the parameters $\theta$ of the prediction network $f_\theta$. Rather than learning a fixed $\theta$, the model learns a generator $H$ that maps task embeddings $z_j$ to task-specific parameters $\theta_j = H(z_j)$. This constitutes a form of amortization—task-specific parameters are not learned independently, but inferred via a shared network.
> In the deterministic case, no sampling is involved and no prior is placed.
> In the latent-variable formulation  a prior is placed over task embeddings $z_j$, which induces a prior over parameters $p(\theta) = p(z) {\vert \frac{dH(z)}{dz} \vert}^{-1}$. **This prior captures variability across tasks, but does not condition on individual covariates**.
>
>
> 4. In _amortized inference_, we can learn a function that produces different posterior parameters for each input. However, such models (e.g., Variational Auto-Encoders) are **trained with a shared, fixed prior** $p(\theta)$ across all inputs and only varying posterior $p(\theta \vert x)$.
>
> **Question 2**:
>
> **ACE and ECE refer to the same metric**, differing only in whether the expectation is taken over the true data distribution (in the case of ECE) or over a finite test sample (in the case of ACE). In practice, evaluations are based on empirical estimates, and thus ACE is a slightly more precise term. That said, the term ECE is commonly used for both (so we have no objection to changing the name).
>
> **Question 3**:
>
> Please see the following table summarizing the average running time of each method across all experiments.
>
> Our method differs from others in that it pre-trains a representation and performs variational inference only on the prediction layer. As a result, for smaller models (e.g., linear regression), our approach can be more computationally expensive due to per-environment optimization. However, **for larger models, our method has an advantage** since competing methods must optimize many more parameters (see the reported times for the CelebA and CIFAR experiments).
>
> | Dataset                               | Hardware         | Unit     | Vids   | DUE    | SNGP   | DUL    |
> |---------------------------------------|------------------|----------|--------|--------|--------|--------|
> | Linear regression                     | 2 CPUs           | seconds  | 67.160 | 39.900 | 54.800 | 16.400 |
> | Logistic regression                   | 2 CPUs           | seconds  | 61.250 | 40.300 | 16.500 | 6.290  |
> | Concrete                              | 2 CPUs           | minutes  | 3.237  | 2.070  | 0.700  | 0.400  |
> | Boston                                | 2 CPUs           | minutes  | 3.112  | 2.017  | 0.717  | 0.267  |
> | Wine                                  | 2 CPUs           | minutes  | 2.523  | 1.583  | 1.567  | 0.895  |
> | CIFAR-C (max across corruptions)      | A100 Cloud GPU   | minutes  | 2.516  | 7.447  | 3.527  | 4.040  |
> | Celeb-A (max across tasks)            | A100 Cloud GPU   | minutes  | 2.420  | 4.417  | 4.220  | 3.318  |

---

> > ### Comment · Reviewer_xBus · 2025-08-01
> > **Answer to Rebuttal**
> >
> > Dear authors,
> > I have carefully read all reviews as well as the author responses. Thank you very much for your thorough answers to my questions.
> > Regarding Question 1, I now understand the difference and novelty much better. The difference regarding the dependence on the individual covariates had not been clear to me before.
> > Regarding Question 2, I am still not quite sure. I had believed the difference between ECE and ACE to be a different one, related to how data is binned and weighted (cf., e.g., Barfoot et al. 2025). I would be grateful if you could clarify your argument to me.
> > Regarding Question 3: Thank you for elaborating. I have no issue with the runtime at all, though I do find it valuable to include such additional information in the Appendix.
> >
> > Reference: Barfoot, T., Garcia-Peraza-Herrera, L. C., Akcay, S., Glocker, B., & Vercauteren, T. (2025). Average Calibration Losses for Reliable Uncertainty in Medical Image Segmentation. arXiv preprint arXiv:2506.03942.

---

> ### Author Response · Authors · 2025-08-01
> **Response to Rebuttal**
>
> Dear reviewer,
>
> Thank you for your thorough feedback and for responding so quickly!
>
> We are glad to hear that the distinction and novelty of our approach are now clearer. As mentioned, we will include a corresponding discussion in the Related Work section of the paper, emphasizing the conditioning on individual covariates.
>
> Following your suggestion, we will also add runtime comparisons to the appendix.
>
> Regarding ACE and ECE, thank you for pointing us to the relevant reference. We now see that there is some inconsistency in terminology across the literature. In any case, the metric reported in our paper (as can be seen in the code attached to the original submission) corresponds to what you referred to as ECE. We will ensure that the paper explicitly defines this metric to avoid any ambiguity.
>
> _Metric computation_: We divide the predicted confidence scores (i.e., the predicted probabilities assigned to the predicted class) into 10 equal-width bins with edges at 0.0, 0.1, 0.2, ..., 1.0. For each bin, we compute:
> - the average predicted probability $\hat{p}$
> - the fraction of samples where the predicted label matches the true label
> We then calculate the absolute difference between the bin's accuracy and its average confidence. This difference is weighted by the proportion of total samples that fall into the bin. We report the sum of these weighted differences across all bins.

---

> ### Comment · Reviewer_xBus · 2025-08-01
> **Response**
>
> Dear authors, thank you very much for your quick response. I fully agree that emphasizing the conditioning on individual covariates in the Related Work section will be helpful, as will the runtime comparisons in the Appendix.
>
> I also appreciate your clarification regarding ECE/ACE. Indeed, this is the metric that I had in mind, and there seems to be some inconsistency in terminology across the literature. So I do think laying this out briefly in the paper does make sense.
>
> My questions have been resolved satisfactorily. I do not have any more questions at this time. I'm planning to adjust my score upwards accordingly.

---

> > ### Author Response · Authors · 2025-08-01
> > **Response**
> >
> > We are very happy to hear. Thank you!

---

### Official Review · Reviewer_WREH · 2025-06-25

**Clarity:** 4
**Significance:** 2
**Originality:** 3
**Rating:** 4
**Confidence:** 3

**Summary:**

The authors proposed a modification in the classic variational inference setting by altering the prior by conditional in the training data as well as in a new test point. Then, they compute the posterior via maximizing the ELBO.

In order to account for the absence of the test point $x^*$, from which the prior is dependent upon, the authors use bootstrap to create a synthetic environment that empirically shows to be similar to a family of unseen shifts. They provide a formal result that states the minimum amount of data needed to ensure these bootstrapping indeed leads to a distribution that is close to the true un-observed one.

Finally for the computing of the posterior, in order to avoid an expensive Montecarlo simulation, using the re-parametrization trick they manage to in a single pass of the data (train and test embedding) compute a the variational distribution from which samples from theta are going to be drawn

**Questions:**

- Is clear that the classifier using your method has a better accuracy, but there is any application for the UQ that can be computed from variational methods?
- I really like the linear regression example, but given the size of frontier models, why is the motivation of these type of work still on NN classifiers?

**Ethical Concerns:**

["NO or VERY MINOR ethics concerns only"]

**Final Justification:**

My comments were minor and more related with the motivation of the work rather than about its technical feat. They answered my concerns and thus I decided to maintain my original opinion about the paper (and henceforth my score as well).

**Paper Formatting Concerns:**

No mayor issues

**Quality:**

4

**Strengths And Weaknesses:**

Weaknesses
- The energy function as a prior are hard to just in the non smooth conditioned marginals of language modeling.
- Their procedure is probably intractable for the training data and NN architectures used in frontier models.
- UQ understood as calibration is hard to interpret in classification settings setting where the true distribution is a point mass, say image classification.
Strengths
- Is a clear an effective extension of a well knwon technique (variatonal inference)
- The method provides robustness against distribution shifts
- They provided a good sample procedure to enable differentiable.

---

> ### Author Rebuttal · Authors · 2025-07-30
>
> Dear reviewer,
>
> Thank you for your review. We address the raised questions and weaknesses below.
>
> - **Smoothness in language models:** The non-smooth conditional marginals in language models are the conditional distributions of the target (e.g., next word) given the context. However, **our model operates in a continuous latent space where the conditional distributions are of the parameters $\theta$ (variational)**.
>
> - **Interpretation of calibration for uncertainty and application**: Calibration, accuracy, and other metrics reported in the paper serve to evaluate how well the model captures the true conditional distribution $p(y \vert x)$. Among these, calibration is often viewed not just as an evaluation metric, but as a goal in itself—reflecting the reliability of the predicted probabilities rather than just the correctness of the most likely label. Intuitively, if for every $\alpha \in [0,1]$, for example $\alpha=0.7$, indeed 70% are positive, it reflects correct uncertainty, that can be used for **decision making, selective prediction, and risk-sensitive applications**.
>
> - **Computation of calibration:** In classification, **calibration is estimated via binning**: predicted probabilities are grouped into discrete intervals (bins), and for each bin, the average predicted probability is compared to the empirical accuracy—that is, the proportion of correctly classified examples within that bin. We follow this standard estimation.
>
> - **NN classifiers vs. regression models:** Our use of regression models was mainly since they allow a simple visualization of the model and the predictive variance. However, the rest of our experiments were on NN architectures (e.g., CIFAR, Celeb-A, UCI datasets). In fact, our method is **agnostic to the architecture and can be applied to any neural network**.

---

> > ### Comment · Reviewer_WREH · 2025-08-08
> >
> > Thanks for the response, my questions have been thoroughly clarified.

---

### Official Review · Reviewer_bQJ5 · 2025-07-03

**Clarity:** 3
**Significance:** 4
**Originality:** 4
**Rating:** 4
**Confidence:** 4

**Summary:**

The authors  introduces an adaptive prior to improve uncertainty estimate.

Adaptive prior is a prior distribution conditioned on input covariates unlike traditional fixed prior.  This was intended to provide a measure the distance ofa test point from the training distribution. To do this, the authors resort to energy model Eq 4.

To estimate the posterior, the paper employs an amortized variational inference. Hence the name of the proposed scheme is VIDS (Variational Inference under Distribution Shifts). An inference network is trained to output the parameters of an approximate posterior distribution, conditioned on both training and test data.

The authors adopt synthetic environments for training since t test data from shifted distributions are often unavailable during training.

**Questions:**

The adaptive prior is conditioned on covariates only but the energy based model incorporates likelihood. It is not clear whether we still call the as prior.

Also, it would be better to show the sensitivity to hyper-parameters for synthetic environments, which eventually decide the performance of the VIDS.

**Ethical Concerns:**

["NO or VERY MINOR ethics concerns only"]

**Final Justification:**

This paper is technically solid but the evaluation is still limited. That's why I could not raise the score to "Accept." I'll keep borderline accept.

**Limitations:**

yes.

**Paper Formatting Concerns:**

The formatting of the paper is quite good.

**Quality:**

3

**Strengths And Weaknesses:**

The authors propose a novel idea; adaptive prior, $ p(θ∣x_{1:n}, x^\star)$. This seems like posterior but does not related with the output (target value). So, it could be considered as a new concept of prior.

Utilizing energy-based model and synthetic environment seems resonable.

Also, the performance comparison with DUE, SNGP, and DUL  are convincing.

The manuscript does not include detailed analysis about computation complexity/cost about variational inference in Section 2.4.

There is limited discussion (including appendix) about the choice of hyper parameters.

---

> ### Author Rebuttal · Authors · 2025-07-30
>
> Dear reviewer,
>
> Thank you for your review. We address the raised questions below.
>
> 1. **The prior is not conditioned on the true outcome $y$. It integrates over all possible values $y$**. For example, in binary classification this would mean $\log p(\theta \vert x, y=0) + \log p(\theta \vert x, y=1)$, regardless of the true unknown $y$.
>
> 2. Aside from standard hyperparameters shared across all methods (e.g., batch size, learning rate), the additional **hyperparameters** in our method are for environment sampling: the number of environments, their size, and the penalty coefficients. We found that performance is insensitive to the number of environments, as long as there are more than a few (e.g., we used 10 in the CIFAR experiments).
> We did observe sensitivity to the penalty coefficient, with consistently better results for smaller values.
> Hence, as the tables in the appendix show, for each type of experiment (UCI datasets, 3 corruptions of CIFAR, 3 attributes in Celeb-A) share the same hyper-parameters.
>
> 3. Please see the attached **running times**:
>
> | Dataset                               | Hardware         | Unit     | Vids   | DUE    | SNGP   | DUL    |
> |---------------------------------------|------------------|----------|--------|--------|--------|--------|
> | Linear regression                     | 2 CPUs           | seconds  | 67.160 | 39.900 | 54.800 | 16.400 |
> | Logistic regression                   | 2 CPUs           | seconds  | 61.250 | 40.300 | 16.500 | 6.290  |
> | Concrete                              | 2 CPUs           | minutes  | 3.237  | 2.070  | 0.700  | 0.400  |
> | Boston                                | 2 CPUs           | minutes  | 3.112  | 2.017  | 0.717  | 0.267  |
> | Wine                                  | 2 CPUs           | minutes  | 2.523  | 1.583  | 1.567  | 0.895  |
> | CIFAR-C (max across corruptions)      | A100 Cloud GPU   | minutes  | 2.516  | 7.447  | 3.527  | 4.040  |
> | Celeb-A (max across tasks)            | A100 Cloud GPU   | minutes  | 2.420  | 4.417  | 4.220  | 3.318  |
>
> Unlike other methods, our approach first pre-trains a representation and then performs variational inference only on the prediction layer. Thus, for large models, competing methods must optimize the full parameter set, and thus ours has an advantage. This is reflected in the reported times for the CelebA and CIFAR experiments. However, for small models our method is likely to be more computationally intensive due to computations across multiple environments.

---

> > ### Comment · Reviewer_bQJ5 · 2025-08-05
> > **Clarification and additional analysis**
> >
> > Thank you for the clarification and runtime time cost.

---

### Official Review · Reviewer_5Fqm · 2025-07-03

**Clarity:** 4
**Significance:** 3
**Originality:** 3
**Rating:** 5
**Confidence:** 3

**Summary:**

The paper proposes variational inference under distribution shifts (VIDS). The main idea is a covariate-conditional adaptive prior that increases uncertainty for test data far from the training distribution. The implementation rests on (1) an energy-based prior that has support over training as well as test covariates, (2) an amortized posterior with test covariates as input and (3) a bootstrap sampling scheme to generate synthetic test covariates for fitting the posterior. On synthetic data, VIDS has good predictive performance and captures the correct heteroscedastic variance structure in- as well as out-of-distribution, outperforming baselines (DUE, SNGP, DUL). On small-scale UCI regression and medium-scale image classification datasets (corrupted CIFAR-10, Celeb-A), VIDS achieves the best accuracy under covariate shift.

**Questions:**

**Runtime**: In appendix E.1 the authors mention run times of ~7 minutes for a UCI experiment, and ~18 minutes for the CIFAR-10 / Celeb-A experiments. Do these run times refer to VIDS, or are those the maxima across all methods? Additionally VIDS appears to involve more hyperparameters than the baselines (predominantly related to the synthetic environments). Does this lead to significantly higher cost during grid search over hyperparameters? More details on tuning complexity would be useful to assess the practical efficiency of VIDS relative to baselines.

**Effectiveness of synthetic environments**: While the use of synthetic environments is theoretically supported (Proposition 3.1), it would strengthen the paper to empirically assess how well the sampled shifts align with actual covariate shifts in test data. If feasible, could the authors include such an analysis, or otherwise discuss limitations of the current simulation strategy?

**Baseline comparisons**: The paper omits some widely used uncertainty estimation methods for neural networks, such as deep ensembles, which may perform well under the studied covariate shifts. Can the authors clarify why such baselines were note included in empirical comparisons?

**Ablations**: The framework integrates several components: the covariate-conditional adaptive prior, amortized inference with specific embedding/predictor architecture, and the environment-level variance penalty. Overall, it is unclear how much each component contributes to performance gains. An ablation study (e.g., disabling the environment-penalty) would help isolate their effects and clarify the drivers of performance. Do the authors have any such results, or plans to include them?

**Minor clarifications**:
- In Tables 3 and 4, the values of $m$ (test environment size) and $J$ (number of environments) are larger for UCI regression than for image classification. Could the authors clarify why this is the case? Does this reflect a need for fewer synthetic covariates in high-dimensional settings?
- Algorithm 1 refers to computing the adaptive prior (Eq 5), which involves a normalization constant. Does the implementation actually compute or approximate this constant, or is it ignored during optimization?
- In Figure 2, it seems the covariate shift is applied only to the second feature. If so, could the authors clarify this in the figure caption?
- Could the authors explain the reason for the large batch sizes used for VIDS (Tables 3 and 4), particularly in relation to other methods?
- In Algorithm 1, the use of $m$ and $M$ appears inconsistent, could this be cleaned up for clarity?
- Typo in line 154: "define" should be "defined".
- Typo in line 471

**Ethical Concerns:**

["NO or VERY MINOR ethics concerns only"]

**Final Justification:**

The rebuttal addressed runtime and hyperparameter concerns with detailed comparisons, clarified the role and limits of synthetic environments, and justified the chosen baselines. The ablation results on the environment penalty provided insight into its contribution to performance. While scalability to very large datasets may remain a practical limitation, the method is novel, well-motivated, and empirically strong under covariate shift.
Recommendation: Accept.

**Limitations:**

yes

**Quality:**

3

**Strengths And Weaknesses:**

### Strengths

**Novelty of adaptive prior**: The paper introduces a covariate-dependent adaptive prior based on an energy objective, an interesting idea in the context of Bayesian neural networks under distribution shift. The concept seems novel, but is related to previous work by Park and Blei (as described in the paper). By conditioning the prior over model parameters on both training and test covariates, the approach improves out-of-distribution uncertainty. The formulation is intuitive and addresses a limitation in current Bayesian neural network models.

**Sound inference scheme**: Sections 2.4 and 2.5 present a rigorous and well-motivated inference scheme for the proposed method. VIDS combines an energy-based adaptive prior with amortized variational inference conditioned on test covariates, allowing posterior uncertainty to adapt dynamically under covariate shift. The variational posterior is parametrized by an inference network that aggregates learned embeddings of the training covariates along with the test covariate, enabling a flexible and scalable approximation. The use of the reparameterization trick, a diagonal Gaussian variational family, and single-sample Monte Carlo estimation together make the optimization procedure computationally tractable.

**Solid empirical results**: VIDS demonstrates consistent empirical performance across synthetic regression and classification problems as well as small- and medium-scale real-world datasets like CIFAR-10-C, Celeb-A, and UCI regressions. VIDS outperforms established baselines from the OOD generalization literature (DUE, SNGP, and DUL) in terms of predictive accuracy on OOD data. The results suggest that the method is practically effective in improving uncertainty estimates under covariate shift.

### Weaknesses

**Runtime**: A potential limitation of the proposed approach may be computational overhead. VIDS requires fitting the variational posterior in each bootstrapped synthetic environment. Within each environment, the method loops over multiple test covariates, computing ELBO objectives and backpropagating through the inference network. While amortization avoids the cost of posterior inference per test point, the nested loop over synthetic environments and test samples introduces substantial computational complexity, particularly for large datasets or when many environments are needed to cover possible covariate shifts. This may limit the scalability of VIDS.

**Effectiveness of synthetic environments**: Another potential limitation lies in the assumption that the distribution of test-time covariates can be approximated at training time. The authors address this via a bootstrapping strategy to simulate covariate shift by constructing synthetic environments, grounded in proposition 3.1 and based on prior work from the OOD generalization literature. While theoretically justified, this approach may not fully capture the nature of real-world distribution shifts.

---

> ### Author Rebuttal · Authors · 2025-07-30
>
> Dear reviewer,
>
> Thank you for dedicating the time to write a thorough review. Please see below.
>
> **1. Runtime:**
>
> The reported run-times were for the entire experiment (all methods). Here is a table summarizing the average running time of each method across all experiments:
>
> | Dataset                               | Hardware         | Unit     | Vids   | DUE    | SNGP   | DUL    |
> |---------------------------------------|------------------|----------|--------|--------|--------|--------|
> | Linear regression                     | 2 CPUs           | seconds  | 67.160 | 39.900 | 54.800 | 16.400 |
> | Logistic regression                   | 2 CPUs           | seconds  | 61.250 | 40.300 | 16.500 | 6.290  |
> | Concrete                              | 2 CPUs           | minutes  | 3.237  | 2.070  | 0.700  | 0.400  |
> | Boston                                | 2 CPUs           | minutes  | 3.112  | 2.017  | 0.717  | 0.267  |
> | Wine                                  | 2 CPUs           | minutes  | 2.523  | 1.583  | 1.567  | 0.895  |
> | CIFAR-C (max across corruptions)      | A100 Cloud GPU   | minutes  | 2.516  | 7.447  | 3.527  | 4.040  |
> | Celeb-A (max across tasks)            | A100 Cloud GPU   | minutes  | 2.420  | 4.417  | 4.220  | 3.318  |
>
> You are correct that using multiple environments increases computational cost. However, this _overhead is significant only for small models_ with few parameters. For larger models (CIFAR, CelebA experiments), our method is more efficient: it pre-trains a representation using standard likelihood optimization and performs variational inference _only on the final prediction layer_. In contrast, _competing methods involve more complex computations across all layers_.
>
> As for **hyperparameter tuning**, our method shares the standard set of hyperparameters used by all methods (e.g., batch size, learning rate). The additional hyperparameters relate to environment sampling: the number of environments, their size, and the penalty coefficients. We found that performance was largely insensitive to the number of environments, as long as there are more than a few (we used 10 in the CIFAR experiments). We did observe sensitivity to the penalty coefficient, with consistently better results for smaller values.
>
> We will include these additional details in the appendix.
>
> **2. Effectiveness of synthetic environments:**
>
> The main limitation of our strategy lies in the support of the source and target distributions. If the supports overlap sufficiently—or, as in our regression experiments, if learning from a sub-region of the training support suffices—then the method is effective.
>
> To build intuition, we can consider the linear regression example. Since the noise variance grows with $x$,  the region near $x=a$ is most informative for learning the heteroscedasticity structure. Given that training inputs are uniformly distributed, the probability that a sample falls within an $\epsilon$--neighborhood of $a$ is $\frac{\epsilon}{a}$. The probability $\rho$ that at least $k$ out of $m$  samples in a single environment fall in that region follows a binomial distribution. To ensure, with confidence at least $1-\alpha$, that at least one of $J$ sampled environments contains $k$ such points requires $J \geq \frac{\log \alpha}{\log 1-\rho}$.
> _For example, with $\alpha=0.05, m=20, k=10, a=0.5, \epsilon=0.1$ this yields a requirement of at least 11 environments._
>
> A similar argument can be developed for all experiments. For example, in the CIFAR-experiments we have a probability of 0.1 of a single sample to be corrupted, and then the same calculation applies only with the corresponding $\rho$.
>
> **3. Baseline comparisons:**
>
> We mention ensemble models in the related work section, as they are commonly interpreted as uncertainty estimators. However, their uncertainty estimates arise from averaging predictions across multiple models—effectively _replacing the original prediction model rather than evaluating uncertainty of a given model_. As a result, the quality of the uncertainty estimate depends heavily on the strength of the ensemble, making it an unfair basis for comparison.
>
> Moreover, ensemble methods do not explicitly address distribution shifts, unlike many distance-based approaches. For these reasons, our comparisons focus on methods that are most closely aligned with our approach: those that integrate Bayesian and distance-based components.
>
> **4. Ablation:**
> We did not include an ablation study since the components of our method are _not designed to function in isolation_. By sampling a sufficiently large number of synthetic environments, we are guaranteed to encounter one that approximates the unseen distribution shift. However, since we don’t know which one, the model must be constrained to perform well across all environments.
>
> That said, _we did run the proposed ablation_. The results below show that without the penalty across environments, _our method still outperforms competing baselines in most cases, but with a smaller margin_ than the complete version.
>
> | Data                    | Metric (Avg.) | DUE   | SNGP  | DUL   | VIDS  | VIDS - no env penalty |
> |-------------------------|---------------|-------|-------|-------|------------------|-------------------------------|
> | Concrete                | RMSE          | 0.490 | 0.532 | 0.484 | 0.441            | 0.660                         |
> | Linear                  | RMSE          | 0.110 | 0.488 | 0.081 | 0.068            | 0.091                         |
> | CIFAR-C: Defocus        | Accuracy      | 0.286 | 0.426 | 0.210 | 0.499            | 0.391                         |
> | CIFAR-C: Contrast       | Accuracy      | 0.291 | 0.437 | 0.098 | 0.525            | 0.435                         |
> | CIFAR-C: Blur           | Accuracy      | 0.395 | 0.494 | 0.495 | 0.635            | 0.530                         |
> | Celeb-A: Gray Hair      | Accuracy      | 0.896 | 0.896 | 0.943 | 0.946            | 0.926                         |
> | Celeb-A: Pale Skin      | Accuracy      | 0.842 | 0.886 | 0.842 | 0.892            | 0.874                         |
> | Celeb-A: Heavy Makeup   | Accuracy      | 0.875 | 0.878 | 0.875 | 0.939            | 0.921                         |
> | Logistic                | Accuracy      | 0.884 | 0.844 | 0.900 | 0.900            | 0.884                         |
> | Wine                    | RMSE          | 0.763 | 0.807 | 0.914 | 0.782            | 0.764                         |
> | Boston                  | RMSE          | 0.962 | 0.934 | 1.165 | 0.938            | 0.834                         |
>
> **5. Minor clarifications:**
> - **Tables 3 and 4:** The main reason lies in the experimental design. For the classification datasets, we had access to additional annotations: corruption types in CIFAR and attributes in CelebA. This allowed us to define distribution shifts explicitly. In contrast, for the regression datasets, the subtypes were constructed via clustering, making the shifts less clearly defined.
> - **Normalization constant:** Thank you for pointing this out. The implementation approximates the log of the normalizing factor $z$ using the log of the mean of exponential log-likelihoods: $\log v \approx  \log (\frac{1}{n} \sum_i exp(v_i))$ where $v_i$ are the integrated log likelihoods. We will add this to the manuscript.
> - **Figure 2:** Yes, the shift is one feature which is constant at training and no longer at test. We will explain this in the legend as suggested (the main text already mentions this).
> - **Batch sizes:** the reported numbers account for the multiple environments.
> - **Notation:** Throughout all the paper $M$ is the number of overall test examples, and $m$ is the number of environment test examples (within a synthetic environment).

---

> > ### Comment · Reviewer_5Fqm · 2025-08-03
> >
> > Thank you for your extensive rebuttal.
> >
> > The runtime in comparison to baselines is much clearer to me now, and I appreciate the additional intuition on the requirements w.r.t. the number of synthetic environments.
> > I acknowledge the deliberate focus of the baselines on distance-based Bayesian methods.
> > The ablation of the environment penalty gives a good impression of the effect of constraining the model to work across several sampled environments.
> >
> > I will keep my recommendation to accept the paper.

---

> > > ### Author Response · Authors · 2025-08-03
> > > **Response**
> > >
> > > Dear reviewer,
> > >
> > > We are happy to hear, and appreciate your response.
> > > Thank you.

---

### Decision · Program_Chairs · 2025-09-17

**Decision:**

Accept (poster)

**Comment:**

This paper considers an uncertainty learning and quantification problem under covariate shift. To this end, the paper proposes a Bayesian framework that leverages an adaptive prior, which is conditioned on both training and new covariates. The paper mainly claims that the proposed adaptive prior is novel, achieving strong empirical results under covariate shift. The following includes discussed strengths and remaining weaknesses:

**Strengths**:
* The proposed adaptive prior is novel.
* The empirical results are stronger than baselines under covariate shift.
* The proposed overall method is sound.

**Weaknesses**:
* The method has scalability and performance issues on very large and complex datasets.

As the novelty is clear, the effectiveness of the adaptive prior is empirically validated, and all reviewers are satisfied with this paper, I agree to accept this paper.